environmental chemistry

electrokinetic remediation, permeable reactive barrier, Cr(VI)-contaminated soil, chemical additives

**Authors for correspondence:**
Binquan Jiao
e-mail: j.binquan@cqu.edu.cn
Ning Lu
e-mail: cqsft_lu@163.com
Dongwei Li
e-mail: litonwei@cqu.edu.cn

†These authors contributed equally to the work.

This article has been edited by the Royal Society of Chemistry, including the commissioning, peer review process and editorial aspects up to the point of acceptance.

# Effect of chemical additives on electrokinetic remediation of Cr-contaminated soil coupled with a permeable reactive barrier

Xu Yu[1],[†], Faheem Muhammad[1],[†], Yujie Yan[1],[†],
Lin Yu[1,2],[†], Huilin Li[1],[†], Xiao Huang[1],[†], Binquan Jiao[1,2],
Ning Lu[3] and Dongwei Li[1]

[1]State Key Laboratory of coal mine disaster dynamics and control, and [2]City College of Science and Technology, Chongqing University, Chongqing, 400044, People's Republic of China
[3]College of Safety Engineering, Chongqing University of Science and Technology, Chongqing, People's Republic of China

DL, 0000-0001-9437-4510

Chromium (Cr) contamination in soil, especially Cr(VI), is a serious threat to the environment and human health. The electrokinetic remediation (EKR) is a promising technology to remediate the Cr(VI). Therefore, in this study, EKR coupled with a permeable reactive barrier (PRB) was used to treat the Cr(VI)-contaminated soil. The CTMAB-Z, a modified zeolite (prepared with cetyltrimethyl ammonium bromide) alone and a mixture of CTMAB-Z and Fe(0) were used as PRB-1 and PRB-2 reactive media, respectively. The effect of chemical enhancers/additives, i.e. DL-tartaric acid and Tween 80 on EKR of Cr(VI) was also analysed in the contrasting experiments. While the effects of repair time, voltage gradient and DL-tartaric acid concentration on Cr(VI) remediation were investigated by using the multifactor orthogonal experiment which was based on contrasting experiments. The contrasting experiment results showed that the highest Cr(VI) removal rate (66.27%) and leaching efficiency (71.29%) were observed in the experimental group which had DL-tartaric acid and PRB-2. Furthermore, the multifactor orthogonal experiment results had depicted that the highest Cr(VI) removal rate (80.92%) and leaching efficiency (85.25%) were achieved after treating the samples at a voltage gradient of $2.5 \, \text{V cm}^{-1}$ for 8 days in the presence of 0.15 M concentration of DL-tartaric acid. This study demonstrated that Cr(VI) remediation through EKR process could be significantly enhanced by the use of PRB and additives.

# 1. Introduction

The chemical substances formed during the rapid industrialization and agricultural production have polluted the soil, which is one of the natural resources used as a growth medium for plant development [1,2]. The polluted soils have a number of heavy metals including Cr(VI). Cr(VI) is one kind of highly carcinogenic and soluble ion which exists as oxyanions, i.e. $CrO_4^{2-}$, $HCrO_4^-$ and $Cr_2O_7^{2-}$. Its exposure to ecosystem via plant uptake, water pollution through leaching and run-off (as a result of rainfall) is a great threat to living biota, especially humans [3,4]. Hence, the remediation of Cr-contaminated soil is necessary for environmental sustainability.

Currently, electrokinetic remediation (EKR) is regarded as one of the feasible remediation technologies. A number of researchers have used EKR for remediation of contaminated water and soils. Basically, this technique's principle is on the migration of heavy metal ions towards the cathode or the anode region under the action of electric field which is supplied through an external voltage source. While the contaminated samples are placed in the sample area situated between the anode and cathode electrodes [5–11]. Moreover, the EK process is also responsible for oxidation ($2H_2O \rightarrow 4H^+ + 4e^- + O_2$) and reduction ($2H_2O + 2e^- \rightarrow H_2 + 2OH^-$) of water molecules at the anode and cathode, respectively [12]. Regarding Cr, Cr(III) is situated in sediment and exists as a cation migrating towards the cathode, while Cr(VI) is an anion migrating towards the anode.

According to the result of early study, the efficiency of EKR with contaminated soil in a short period of time was poor because the heavy metals existed as precipitates in the soil. Later on, different chemical additives/enhancers were introduced to the process to enhance the removal efficiency of heavy metals. The mechanisms included solubilization of pollutants through acidolysis and complexation effect which, in turn, enhance the migration capacity of pollutants, changing the direction of electro-osmotic flow and intensity of current, controlling the pH of electrolyte, etc. [13–18]. These chemical enhancers were categorized as alkalis and their salts, organic and inorganic acids, chelating agents, surfactants, redox agents and buffer solutions. For example, citric acid was used as an additive to improve the removal efficiency of Cr from contaminated soil in the experiments conducted by Fu et al. [12] and Wu et al. [19]. Tang et al. also reported the effective remediation of the heavy metals (Cu, Zn, Cr, Pb, Ni and Mn) from sludge by adding GLDA and rhamnolipid [20]. Similarly, the remediation efficiency of dredged sediments polluted with heavy metals (Cu, Zn, Cd, Cr, Pb and Ni) was enhanced with the addition of citric acid, acetic acid, humic acid, etc. [21].

Recently, the permeable reactive barrier (PRB) technology was introduced and coupled with the EKR process, as it can effectively adsorb several pollutants. The number of materials used as reactive media in PRB technology includes zero-valent iron, activated carbon, zeolite, humic acid, etc. [22–26]. The contaminated materials can be adsorbed or precipitated by physical, chemical and biological means upon passing through PRB. Compared with simple EKR, the EKR coupled with PRB technology brings better efficiency and has been widely used for remediation of heavy metal-contaminated soils. Xue et al. successfully remediated the Cr-contaminated soil by using zeolite and zero-valent iron as a reactive media in PRB [27]. Zhang et al. effectively removed the Cr(VI) after using a calcined-hydrotalcite-based reactive material for treatment of Cr(VI)-contaminated soil [28]. Similarly, Weng et al. treated the hyper-Cr(VI) contaminated clay by using zero-valent iron as PRB reactive media, and satisfactory results were achieved [25]. Among these reactive materials, zeolite has a microporous structure composed of silicon tetrahedral and aluminium octahedral units. Besides the high specific surface area, its porous structure can adsorb high amount of heavy metal. Generally, heavy metal cations are adsorbed on the surface of zeolite by ion exchange mechanism, as the silica-alumina oxygen units of zeolite caused a negative charge on the surface of zeolite [29–32].

Although zeolite is widely used as a PRB material, its adsorption capacity of Cr(VI) anions is limited by its native surface characteristics. This negatively charged zeolite can be further modified with cationic surfactants to form micelle-like bilayer structures, changing the zeta potentials, therefore enhancing the adsorbing capacity of anions [33–35]. To further improve the adsorption capacity, the zeolite in this study was modified with cetyltrimethyl ammonium bromide (CTMAB). During the EKR experiment, the PRB was classified as PRB-1 and PRB-2 depending upon reactive media. The PRB-1 possessed the modified zeolite (CTMAB-Z) as adsorbate and PRB-2 had composite adsorbate, i.e. CTMAB-Z and Fe(0). Moreover, the effect of additives (DL-tartaric acid and Tween 80) on the Cr(VI) remediation was also investigated.

# 2. Material and methods

## 2.1. Materials and chemicals

The zeolite was purchased from Sinopharm Group Chemical Reagent Co. Ltd (Shanghai, China). The Cr-contaminated soil samples were collected from an industrial park situated in Chongqing, China, dried and sieved through a 100-mesh sieve. Deionized water was used throughout the experiment and all chemicals were of analytical grade.

## 2.2. Preparation of modified zeolite (CTMAB-Z)

The zeolite (20–40 mesh) was dried at 105° for 12 h after several times of washing with deionized water. Subsequently, the zeolite was mixed with 0.1 M cetyltrimethyl ammonium bromide (CTMAB) at a ratio of 1 : 5 (solid-to-liquid, g ml$^{-1}$), then shaken at a rate of 150 r.p.m. for 6 h at room temperature. Finally, the dried zeolite was used as PRB reactive media after 8–10 times of washing with deionized water.

## 2.3. Experimental device

The EKR experiments were carried out in a self-made rectangular glass reactor. The equipment consisted of three main parts: sampling region/soil remediation chamber, cathode compartment (C1) and anode compartment (C2). The soil remediation chamber ($100 \times 70 \times 80$ mm) was evenly divided into S1, S2 and S3 regions directed from the cathode to the anode. The power was supplied through low-voltage DC power source which was connected to the graphite anode and the stainless-steel cathode via an aluminium wire. But, the 0.1 M potassium chloride solution was used as the initial electrolytic solution in the electrolytic cell. All the EKR experiments were performed with 120 g of soil. Moreover, the PRB was placed between the S3 region and the C2 region as shown in figure 1.

## 2.4. Design of experiment

### 2.4.1. Equilibrium adsorption experiments

The equilibrium adsorption experiment was carried out in 100 ml conical flasks. The specified amounts of adsorbents (the modified zeolite (CTMAB-Z) and original zeolite (UM-Z)) were added into the conical flask which already contained 20 ml of $K_2Cr_2O_7$ solution with determined concentrations. The solution was filtered after shaking at room temperature for 6 h at a speed of 150 r.p.m. The concentration of Cr(VI) in supernatant solution was determined and the removal rate was calculated as follows:

$$\partial = \frac{\partial_0 - \partial_e}{\partial_0} \times 100,$$

where $\partial_0$ and $\partial_e$ are the Cr(VI) concentrations (mg l$^{-1}$) before and after the equilibrium adsorption experiment, respectively.

### 2.4.2. Electrokinetic remediation experiment

The contrast experiments were classified into six groups based on experimental conditions and the results are shown in table 1. The PRB was classified as PRB-1 and PRB-2 depending upon reactive media. In PRB-1, only single adsorbent (CTMAB-Z) was used. But, a mixture of CTMAB-Z and Fe(0) was used as a composite adsorbent in PRB-2. In addition, different chemical additives, i.e. 0.1 M DL-tartaric acid, 0.1% Tween 80 and mixture (0.1 M DL-tartaric acid and 0.1% Tween 80), were used as the catholyte to enhance EKR process, it was believed that catholyte can improve the remediation effect by controlling pH and enhancing migration effect [18,36–38]. The samples were remediated for 5 d under a constant supply of voltage (2.0 V cm$^{-1}$) in six experimental groups.

Furthermore, the multifactor orthogonal experiments based on the results of these six experimental groups were carried out to further enhance the Cr(VI) remediation efficiency. The effects of three different factors (repair time, voltage gradient and concentration of DL-tartaric acid), each with three levels on Cr(VI) remediation efficiency were evaluated in the multifactor orthogonal experiment. The details about the orthogonal experiments are presented in tables 2 and 3.

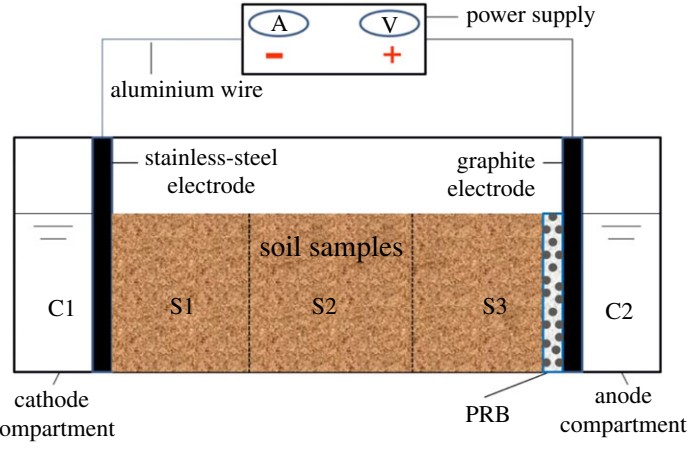

**Figure 1.** The schematic diagram of the EKR experimental device.

**Table 1.** The contrast experimental groups.

| group | cathode compartment | remediation chamber | anode compartment |
|---|---|---|---|
| A1 | KCl | soil | KCl |
| A2 | KCl | soil + PRB-1 | KCl |
| A3 | KCl | soil + PRB-2 | KCl |
| A4 | DL-tartaric acid | soil + PRB-2 | KCl |
| A5 | Tween 80 | soil + PRB-2 | KCl |
| A6 | mixture | soil + PRB-2 | KCl |

**Table 2.** Factors and their levels in the orthogonal experiments.

| | levels | | |
|---|---|---|---|
| factor | 1 | 2 | 3 |
| repair time (A) | 4 d | 6 d | 8 d |
| voltage gradient (B) | 1.5 V cm$^{-1}$ | 2 V cm$^{-1}$ | 2.5 V cm$^{-1}$ |
| DL-tartaric acid (C) | 0.05 M | 0.1 M | 0.15 M |

Finally, two replicates of experiments were carried out to verify the optimal combination obtained by the range analysis in orthogonal experiments.

## 2.5. Experimental calculations

The concentration of Cr(VI) in soil was determined by the ultraviolet spectrophotometer after alkali digestion in accordance with the Chinese standard HJ687-2014 solid waste—determination of Cr(VI). The main processes were as follows: the soil was mixed with $Na_2CO_3/NaOH$ and $K_2HPO_3/KH_2PO_3$ solutions and stirred at 90–95°C for 1 h; After that, the mixture was filtered through the filter pump; in the end, the extracted solution was adjusted to pH $9.0 \pm 0.2$ with $HNO_3$ and fixed to 100 ml before the following testing.

The leaching toxicity of solid waste refers to the concentration of pollutants in the leaching solution of solid waste obtained through a leaching procedure. The leaching toxicity of Cr(VI) in soil was analysed through the ultraviolet spectrophotometer after treating with an acid solution in accordance with the Chinese standard HJ/T299-2007 solid waste leaching toxicity—sulfuric acid and nitric acid method. The soil was mixed with acid solution (sulfuric acid and nitric acid, pH $= 3.20 \pm 0.05$) and shaken at room temperature for $18 \pm 2$ h at a speed of 120 r.p.m., then concentration of Cr(VI) in the

**Table 3.** The orthogonal experimental layout.

| group | A | B | C |
|-------|---|---|---|
| B1 | 1 | 1 | 1 |
| B2 | 1 | 2 | 2 |
| B3 | 1 | 3 | 3 |
| B4 | 2 | 1 | 2 |
| B5 | 2 | 2 | 3 |
| B6 | 2 | 3 | 1 |
| B7 | 3 | 1 | 3 |
| B8 | 3 | 2 | 1 |
| B9 | 3 | 3 | 2 |

**Table 4.** Elemental composition (%) of soil samples.

| O | Ca | Si | Cr | Fe | Ni | Al | Mg | Na | S | Zn |
|---|----|----|----|----|----|----|----|----|----|----|
| 38.04 | 12.38 | 9.26 | 7.73 | 6.58 | 5.88 | 4.32 | 3.86 | 3.73 | 2.47 | 1.79 |
| P | Cu | K | Ti | Cl | Sr | Ba | Pb | Sn | Zr | total |
| 1.42 | 0.75 | 0.55 | 0.48 | 0.41 | 0.11 | 0.11 | 0.05 | 0.04 | 0.04 | 100 |

supernatant after filtering was measured. The Cr(VI) removal rate or leaching efficiency was calculated using the following formula:

$$\gamma = \frac{\gamma_0 - \gamma_t}{\gamma_0} \times 100,$$

where $\gamma$ is the Cr(VI) removal rate or leaching efficiency, $\gamma_0$ is the initial Cr(VI) concentration or leaching toxicity and $\gamma_t$ is the Cr(VI) concentration or leaching toxicity after the experiment.

# 3. Results

## 3.1. Characterization analysis of zeolite before and after modification

The characteristics of soil used in this study were first analysed. The elemental composition of soil was detected with X-ray fluorescence and is shown in table 4; The physical and chemical analyses are mentioned in table 5, while the particle-size distribution of soil samples was also obtained with particle size analyser and is shown in figure 2. After that, the zeolite was also characterized as follows:

(a) *SEM and specific surface area*. The specific surface area has great impact on the adsorption properties of adsorbent material. According to the data of table 6, the specific surface area of UM-Z was higher than that of CTMAB-Z. During the modification with CTMAB, N-terminus adsorption on the surface of zeolite caused the formation of micelle-like cationic surfactant membrane, decreasing the specific surface area. But, the CTMAB-Z possessed higher pore volume and size due to the removal of impurities after washing. These results were consistent with SEM images (figure 3): the CTMAB-Z had a smooth surface when compared with UM-Z. Moreover, a large number of small granular particles were shown in SEM images (figure 3a,b) of UM-Z, indicating the higher specific surface area.

(b) *FTIR analysis*. FTIR analysis is regarded as an effective method for determination of structural characteristics and functional groups of inorganic, organic and compound molecules [23]. The FTIR spectra of UM-Z and CTMAB-Z are shown in figure 4. The new adsorption peaks corresponding to $-CH_2$ were observed around 2918 and 2850 $cm^{-1}$ bands in CTMAB-Z. The antisymmetric and symmetric stretching vibrations around 2918 and 2850 $cm^{-1}$ peaks were corresponding to $-CH_2$. But the plane-shearing vibrational band of $-CH_2$ was observed at a wavelength of 1473 and

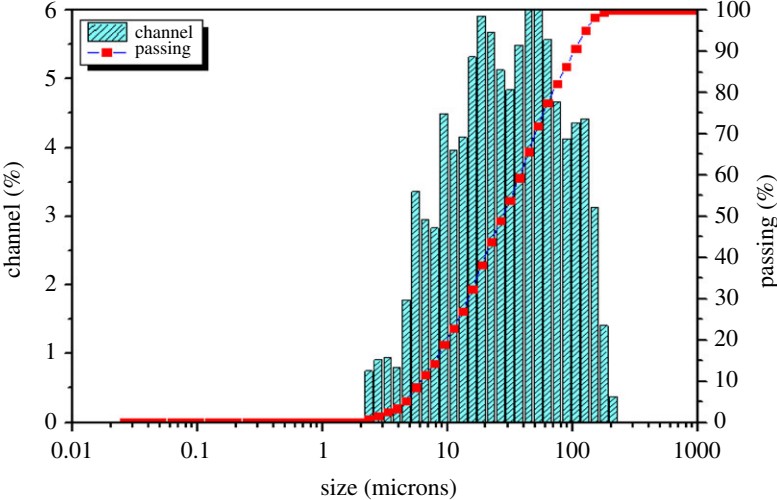

**Figure 2.** The particle-size distribution of soil samples.

**Table 5.** The physical and chemical analysis of soil samples.

| analysis | concentration | method |
|---|---|---|
| pH of soil | 8.14 | 1 : 3 soil/water |
| Cr(VI) concentration | 504 mg kg$^{-1}$ | alkaline digestion |
| leaching concentration | 12.55 μg ml$^{-1}$ | sulfuric acid and nitric acid |

1463 cm$^{-1}$. Moreover, the antisymmetric angular vibrational peak of $-CH_3$ linked to $N^+$ was analysed at 1487 cm$^{-1}$. These peaks along with ammonium peak (1384 cm$^{-1}$) were not observed in UM-Z. It indicated that these peaks resulted due to the successful loading of surfactant onto the zeolite in CTMAB-Z.

(c) *Zeta potential analysis.* The zeta potential is a potential on the sliding surface of colloidal double electric layer which defines the electrostatic properties of the colloid. The electrostatic principle states that positive and negative zeta potential can attract anions and cations, respectively. According to figure 5, the UM-Z had $-49.4$ mV zeta potential. But surfactant (CTMAB) adsorption caused the reverse potential of zeolite (23.6 mv) upon its adsorption on zeolite surface, indicating that CTMAB-Z had a higher Cr(VI) adsorption capability.

## 3.2. Equilibrium adsorption experiment results

### 3.2.1. Effect of Cr(VI) concentration

In figure 6, the Cr(VI) removal rate decreased with an increase of Cr(VI) concentration in both UM-Z and CTMAB-Z, as the adsorbent easily attained the equilibrium condition at higher Cr(VI) concentrations. But, higher removal rate of Cr(VI) was observed in CTMAB-Z when compared with UM-Z at all Cr(VI) levels which was observed by the appearance of both materials (figure 7). The low removal rate of UM-Z was due to the negatively charged silica-alumina oxygen skeleton. By contrast, the modification of zeolite with CTMAB had developed net positive charge on its structure, which was feasible for removal of Cr(VI) anions.

### 3.2.2. Effect of adsorbent dosage

The removal efficiency of different adsorbent dosage on initial Cr(VI) concentration (50 mg l$^{-1}$) is shown in figure 8. Generally, the removal rate increased by increasing the dosage of adsorbent in both cases, i.e. UM-Z and CTMAB-Z. The linear increase in removal rate was observed in UM-Z, but there was no

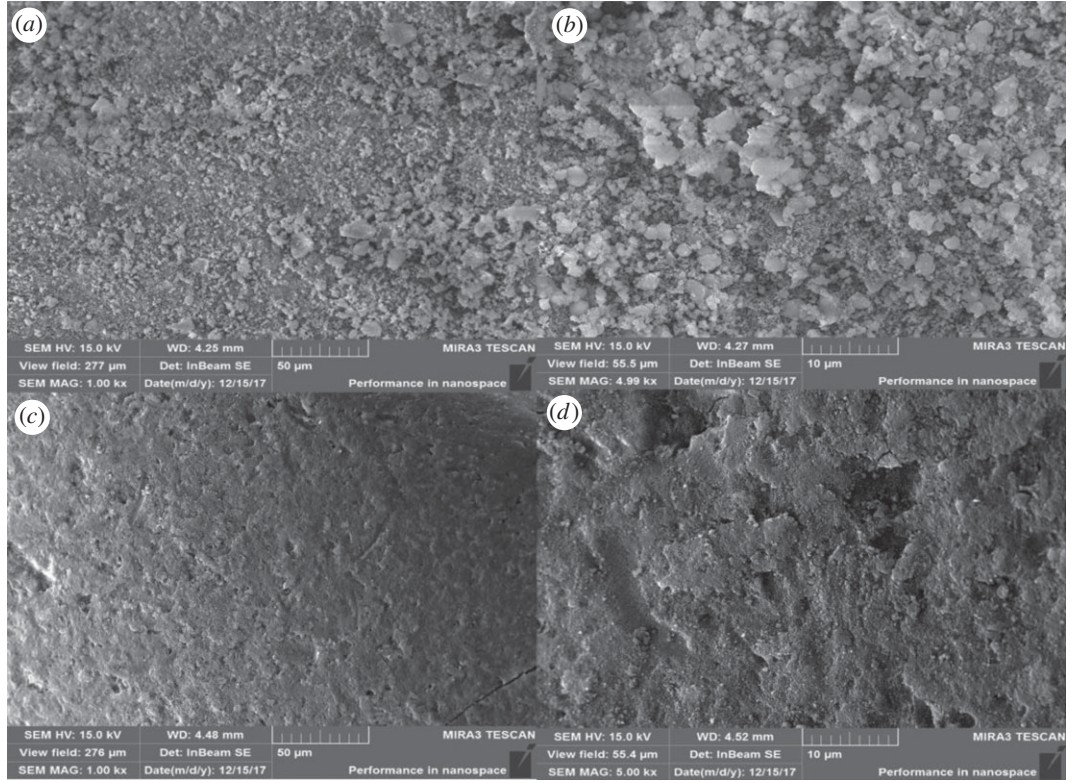

**Figure 3.** SEM images of (*a,b*) UM-Z and (*c,d*) CTMAB-Z.

**Table 6.** Specific surface area analysis.

| zeolite category | specific surface area (m$^2$ kg$^{-1}$) | pore volume (cm$^3$ kg$^{-1}$) | pore diameter (Å) |
|---|---|---|---|
| UM-Z | 32.036 | 0.092 | 114.527 |
| CTMAB-Z | 30.523 | 0.105 | 137.956 |

significant difference among different dosages. By contrast, the removal rate significantly increased up to a certain addition (1 g) of adsorbent dosage and the removal process got slower with further increase in the dosage of CTMAB-Z.

## 3.3. Electrokinetic remediation experiments

### 3.3.1. Macro-phenomena of electrokinetic remediation

Some phenomena observed frequently during the EKR experiments were summarized as follows:

(1) The bubbling phenomenon was observed in cathode and anode regions which was increased with an increment of voltage gradient, indicating the electrolytic reactions were more intense at higher voltage gradient. The specific reactions occurred at cathode and anode regions are given below:

$$\text{cathodic reaction: } 2H_2O + 2e^- \rightarrow 2OH^- + H_2 \uparrow$$
$$\text{anodic reaction: } H_2O - 2e^- \rightarrow 2H^+ + \frac{1}{2}O_2 \uparrow$$

(2) The electrolytic solution in the anode compartment became darker with the passage of time, due to the migration of Cr(VI) from the cathode to the anode under the action of the electric field during EKR.
(3) The gradual change of colour in the PRB region indicated that Cr(VI) had been successfully adsorbed on reactive media.

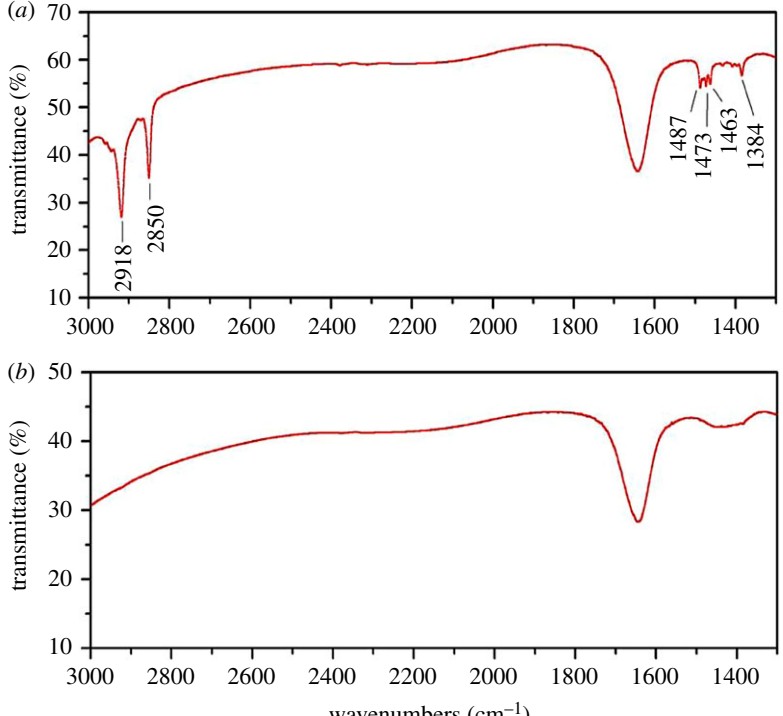

**Figure 4.** FTIR spectra of (*a*) CTMAB-Z and (*b*) UM-Z.

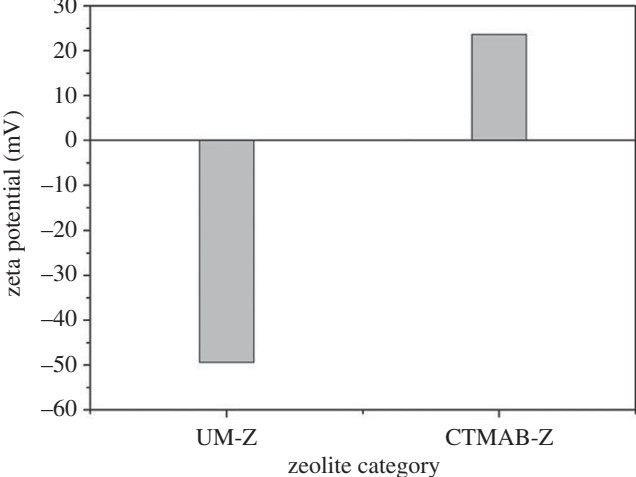

**Figure 5.** Zeta potential of UM-Z and CTMAB-Z.

(4) The soil became hardened with the passage of time. Both the precipitation of heavy metal ions in alkaline environment and the volatilization of soil moisture caused by thermal effect led to this phenomenon.

### 3.3.2. The contrast experiments

#### 3.3.2.1. The current variations with respect to time

The similar trend regarding current variations was observed in all experimental groups: initially, the current increased and then gradually decreased (figure 9). The current was proportional to the number of free ions, i.e. $OH^-$ and $H^+$, coming from the electrolysis reaction and the soluble salts and heavy metal ions released from the soil [27]. Therefore, the increase of current was due to more free ions and continuous release of metal ions from soil [22]. But the decrease of current was the result of two mechanisms: first, under the action of an electric field, heavy metals and other free ions migrated

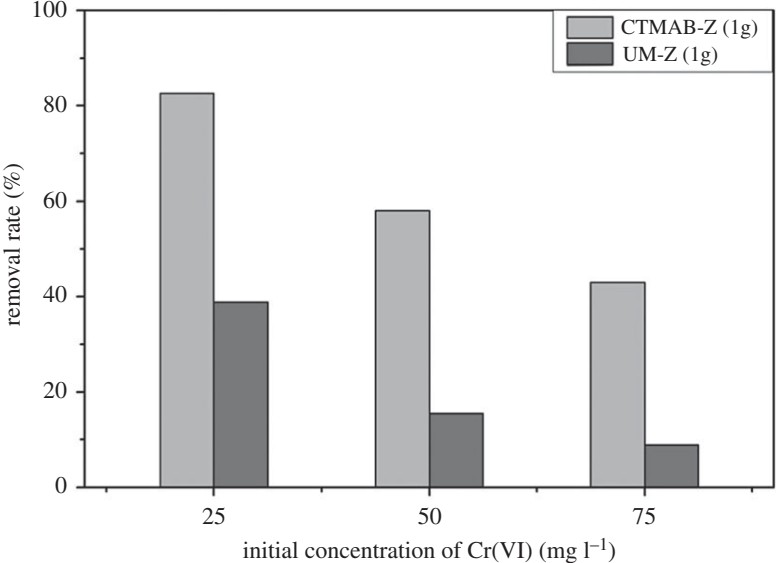

**Figure 6.** Effect of Cr(VI) concentration.

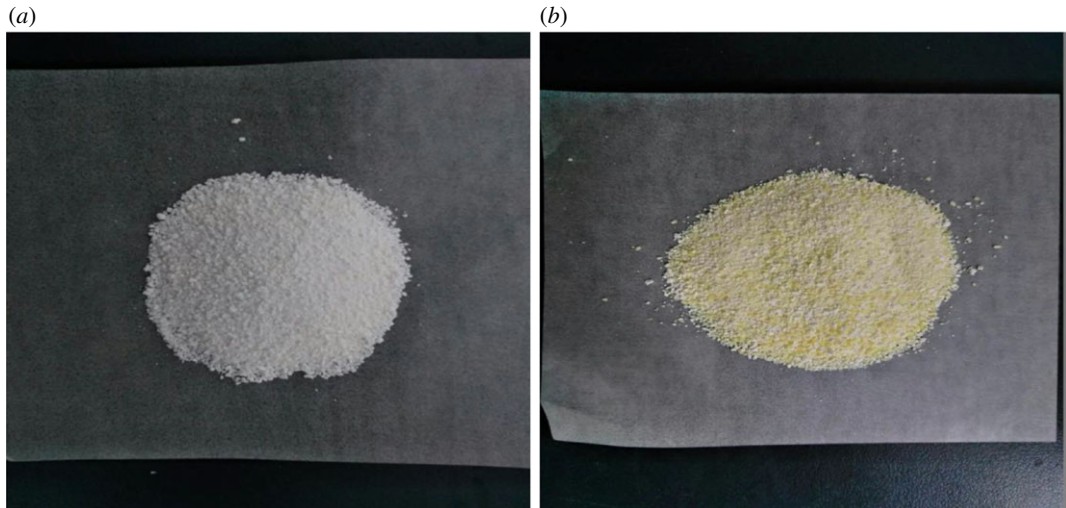

**Figure 7.** UM-Z (*a*) and CTMAB-Z (*b*) after adsorbing Cr(VI).

towards their respective electrode compartments during the EKR; then, the precipitation of $OH^-$ with metal ions in the alkaline environment caused an increase in resistance.

Overall, the higher current was observed in experimental groups which only have KCl electrolytic solution as compared to those groups which also had DL-tartaric acid or Tween 80 additives. Theoretically, KCl can be easily disassociated into free ions, increasing the ion concentration and also the solution conductivity. But the lack of additives resulted in a quick decrease in current with the passage of time. Comparatively, the lowest current was observed in the A5 group (having 0.1% Tween 80), and the low ion concentration in the system was due to less concentration of ions dissolved from the Tween 80. According to figure 9, the highest current was observed in the group A2 (CTMAB-Z) when compared with A1 and A3, indicating that the CTMAB-Z favoured the release of adsorbed ions from the soil due to a concentration gradient. The lower current in the group A3 than the A2 group was due to the release of Fe(II) in the acidic environment. In the later stage, the Fe(II) was oxidized to Fe(III) with the decrease of Cr(VI) to Cr(III). Part of these Fe(III) and Cr(III) ions was adsorbed on the surface of zeolite, and the remainder can migrate into soil and precipitate, which hindered the migration of free ions.

### 3.3.2.2. pH variations

The pH variations before and after the EKR experiment were analysed and shown are in figure 10. Before EKR, the soil samples acidified with DL-tartaric acid had lower pH when compared with other groups

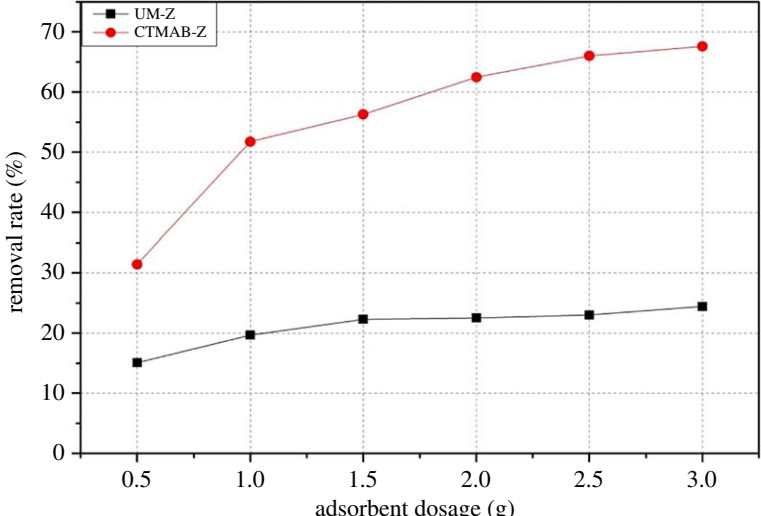

**Figure 8.** Effect of adsorbent dosage.

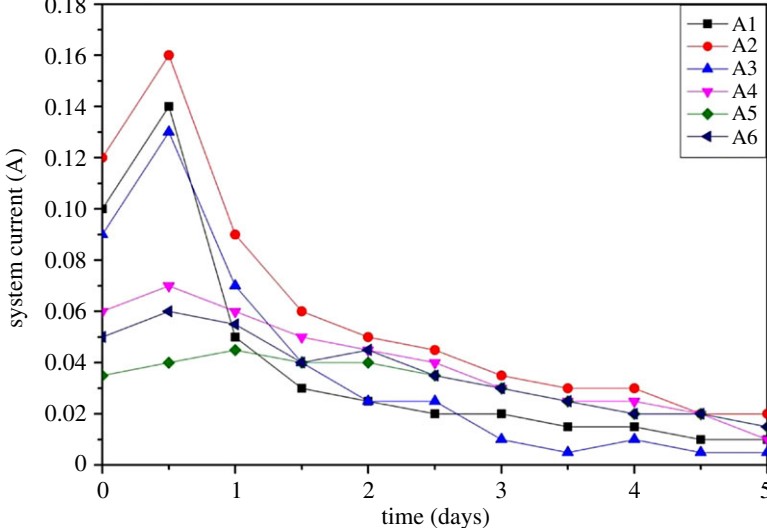

**Figure 9.** The current variations with respect to time.

which possessed pH higher than 7. By contrast, a decrease in pH was observed from the cathode to the anode after EKR. It was due to the generation of $OH^-$ ions at the cathode and $H^+$ ions at the anode as a result of electrolysis of water molecules. In the S1 and S3 regions, the pH was observed in the range of 12–14 and 3–7, respectively. In spite of higher migration rate of $H^+$ than $OH^-$, the native alkaline environment of soil with strong buffering capacity caused the partially alkaline pH of the S2 region. In this study, the addition of PRB resulted in an increase in pH value in the S3 region, as the PRB exhibited a positive surface potential, adsorbing $OH^-$ ions and preventing the migration of $H^+$.

### 3.3.2.3. Remediation efficiency of Cr(VI)

The average removal rate and leaching efficiency of six experimental groups are shown in table 7. The significant differences regarding Cr(VI) removal were observed among these groups. The Cr(VI) remediation efficiency was observed in the following order: A4 > A5 > A6 > A3 > A2 > A1. These results indicated the DL-tartaric acid and PRB-2 improved the Cr(VI) remediation efficiency. Basically, DL-tartaric acid had controlled the pH and accelerated the Cr(VI) migration by acidification effect. Regarding PRB-2, CTMAB-Z and Fe(0) reduced the Cr(VI) concentration through adsorption and reduction effect.

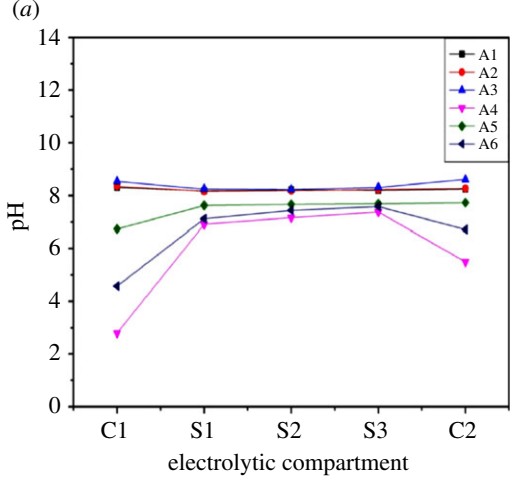

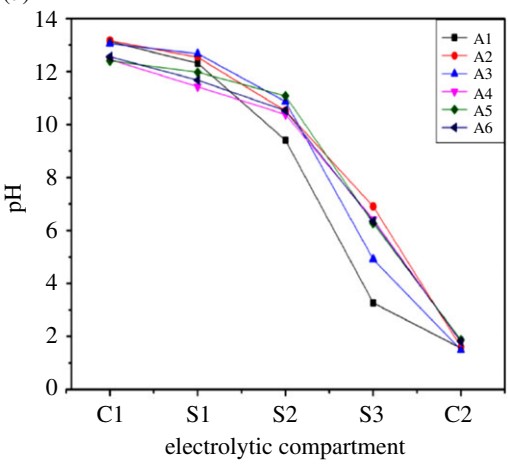

**Figure 10.** pH variations of (*a*) before EKR and (*b*) after EKR.

**Table 7.** Remediation efficiency of Cr(VI) in the contrast experiments.

| group | average removal rate (%) | average leaching efficiency (%) |
|---|---|---|
| A1 | 9.83 | 0.61 |
| A2 | 26.18 | 0.30 |
| A3 | 28.29 | 16.87 |
| A4 | 66.27 | 71.29 |
| A5 | 58.10 | 62.17 |
| A6 | 56.74 | 58.52 |

In this study, A1–A3 groups had no additives, while A4–A6 groups had additives. The groups without additives had better removal rate when compared with leaching efficiency. It might be because the oxidizable and reducible states of Cr(VI) were transformed into the acid extractable state during the EKR process, resulting in a low leaching efficiency. But the removal rate and leaching efficiency were increased after the addition of additives, which indicated that these additives had good effect during the EKR process. Overall, the A2 and A3 groups had higher remediation efficiency when compared with the A1 group due to the presence of PRB. As mentioned earlier, the CTMAB-Z in PRB-1 possessed adsorbing effect and the addition of Fe(0) in PRB-2 further brought the reduction effect. As a result, the A2 and A3 groups exhibited higher remediation efficiencies than the A1 group.

According to figure 11, the Cr(VI) remediation efficiency gradually decreased from the S1 region to the S3 region due to the migration of Cr(VI) from the cathode to the anode. The lower pH also contributed to a higher adsorption of Cr(VI), therefore lowering the remediation efficiency in the S3 region. The results were in accord with the report of Al-Hamdan [5]. In the case of the group A3, the remediation efficiency of Cr(VI) increased from the S1 to the S3 region due to the increase of $H^+$ ion concentration. This increase of $H^+$ ion concentration caused the oxidation of Fe(0) to Fe(II) ions, along with the reduction of Cr(VI) to Cr(III).

### 3.3.3. The multifactor orthogonal experiments

The main purpose of the orthogonal experiment was to achieve the optimum levels of concerned factors, i.e. repairing time, voltage gradient and DL-tartaric acid concentration.

(a) *The current variations with respect to time.* Among all groups, a similar trend (initial increase and then gradual decrease) regarding current changes was analysed and presented in figure 12. The current variations were proportional to the concentration of free ions, and further explanations can be seen in §3.3.2.1. Moreover, the current was also mainly affected by voltage gradient: the higher the voltage gradient, the higher the current.

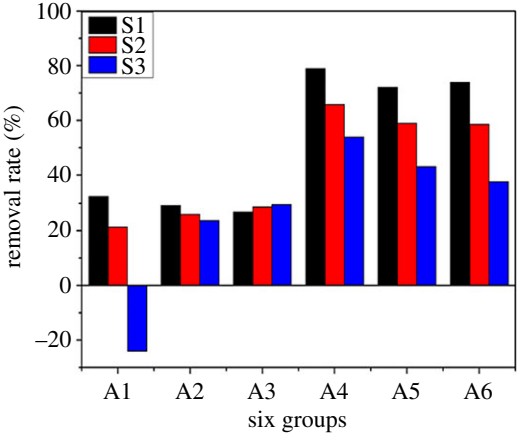
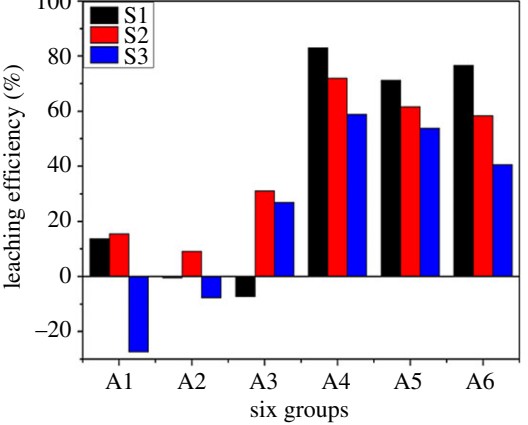

**Figure 11.** Remediation efficiency of Cr(VI) in three regions.

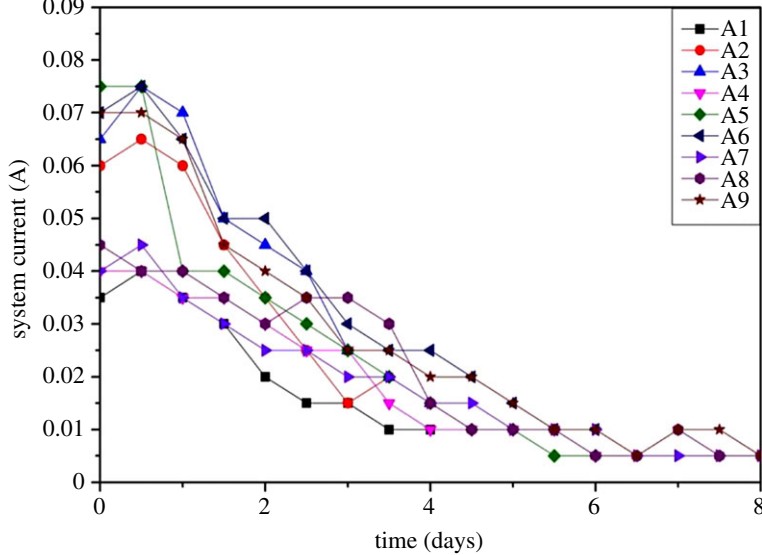

**Figure 12.** The current variations with respect to time.

(b) *Remediation efficiency of Cr(VI)*. During orthogonal experiment, the highest average removal rate (78.74%) and leaching efficiency (82.65%) of Cr(VI) were observed in the group B9 (table 8) which was achieved after preparing the sample with 0.15 M of DL-tartaric acid at voltage gradient of 2.5 V cm$^{-1}$ for 8 days.

The influence of various factors on the remediation efficiency of Cr(VI) is shown in figure 13. First of all, the highest remediation efficiency of Cr(VI) was observed after treating the samples for 8 days, indicating that increasing the period of time benefited the migration of Cr(VI). Then, the highest voltage gradient (2.5 V cm$^{-1}$) also caused the highest migration speed and the highest removal rate. Moreover, increasing the concentration of DL-tartaric acid also caused a higher release of Cr(VI), and the best results were obtained with 0.15 M DL-tartaric acid. In addition, according to $R$ values (table 9), these factors had varying degrees of influence on the remediation efficiency of Cr(VI) and the order is as follows: repair time (A)>voltage gradient (B)>concentration of DL-tartaric acid (C). Overall, the optimum level of these factors regarding remediation efficiency of Cr(VI) in this study is summarized as A3 (8 days repair time), B3 (2.5 V cm$^{-1}$ of voltage gradient) and C3 (0.15 M DL-tartaric acid concentration). In other words, the highest remediation efficiency of Cr(VI) was achieved after treating the samples at a voltage gradient of 2.5 V cm$^{-1}$ for 8 days in the presence of 0.15 M concentration of DL-tartaric acid.

Furthermore, the EKR based on the optimum level was carried out. For this purpose, the two replicates of experiments were carried out in accordance with this optimum combination (A3B3C3). In

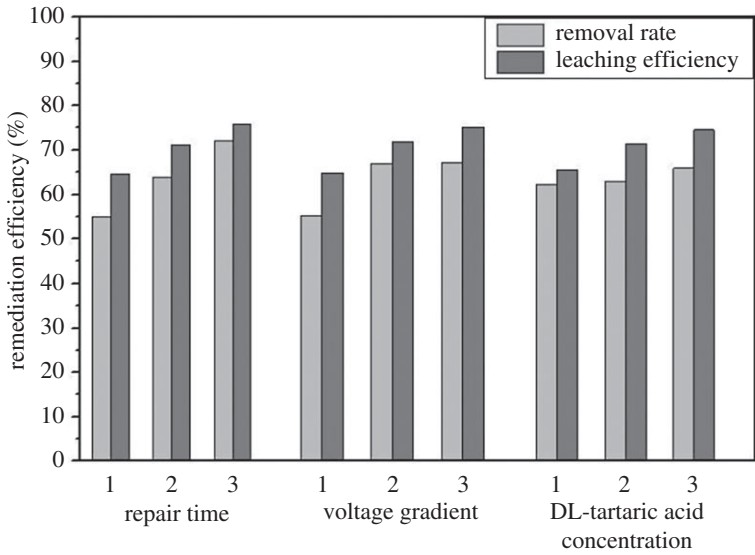

**Figure 13.** The effect of experimental factors on remediation efficiency of Cr(VI).

**Table 8.** Remediation efficiency of Cr(VI) in the orthogonal experiments.

| group | A | B | C | removal rate (%) | leaching efficiency (%) |
|---|---|---|---|---|---|
| B1 | 1 | 1 | 1 | 50.74 | 55.33 |
| B2 | 1 | 2 | 2 | 56.87 | 68.30 |
| B3 | 1 | 3 | 3 | 57.52 | 70.35 |
| B4 | 2 | 1 | 2 | 53.15 | 63.52 |
| B5 | 2 | 2 | 3 | 73.27 | 77.64 |
| B6 | 2 | 3 | 1 | 65.40 | 72.18 |
| B7 | 3 | 1 | 3 | 66.93 | 75.59 |
| B8 | 3 | 2 | 1 | 70.65 | 69.22 |
| B9 | 3 | 3 | 2 | 78.74 | 82.65 |

**Table 9.** Range analysis of Cr(VI).

| | removal rate | | | leaching efficiency | | |
|---|---|---|---|---|---|---|
| $K_{j1}$ | 165.13 | 170.82 | 186.79 | 193.98 | 194.44 | 196.73 |
| $K_{j2}$ | 191.82 | 200.79 | 188.76 | 213.34 | 215.16 | 214.47 |
| $K_{j3}$ | 216.32 | 201.66 | 197.72 | 227.46 | 225.18 | 223.58 |
| $k_{j1}$ | 55.04 | 56.94 | 62.26 | 64.66 | 64.81 | 65.58 |
| $k_{j2}$ | 63.94 | 66.93 | 62.92 | 71.11 | 71.72 | 71.49 |
| $k_{j3}$ | 72.11 | 67.22 | 65.91 | 75.82 | 75.06 | 74.53 |
| optimal levels | A3 | B3 | C3 | A3 | B3 | C3 |
| $R$ | 17.07 | 10.28 | 3.65 | 11.16 | 10.25 | 8.95 |
| main sequence | A > B > C | | | A > B > C | | |

table 10, the average removal rate and leaching efficiency of Cr(VI) were 80.92% and 85.25%, respectively. In addition, the remediation efficiency of S3 was highest and it was significantly improved in this section when compared with contrast experiment. These results also indicated that Fe(0) had a good reduction

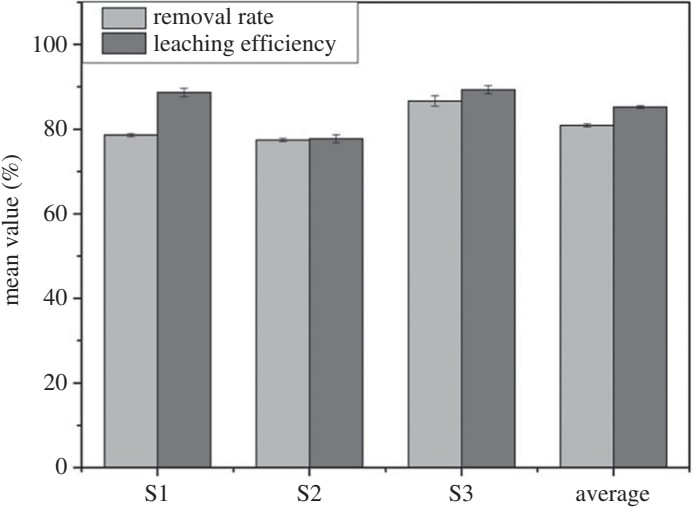

**Figure 14.** The error analysis.

**Table 10.** Remediation efficiency of Cr(VI) in the optimal combination experiments.

| group | removal rate | | | | leaching efficiency | | | |
|---|---|---|---|---|---|---|---|---|
| | S1 (%) | S2 (%) | S3 (%) | average (%) | S1 (%) | S2 (%) | S3 (%) | average (%) |
| C1 | 78.33 | 77.75 | 87.54 | 81.21 | 89.34 | 77.07 | 90.02 | 85.48 |
| C2 | 78.90 | 77.18 | 85.82 | 80.63 | 87.98 | 78.43 | 88.66 | 85.02 |
| average | 78.62 | 77.46 | 86.68 | 80.92 | 88.66 | 77.75 | 89.34 | 85.25 |

effect in the presence of optimum experimental conditions. The error analysis of the experiments is shown in figure 14.

# 4. Discussion

## 4.1. Removal mechanisms of Cr(VI)

The EKR is based on the migration of heavy metal ions towards the cathode or the anode region under the action of an electric field which was supplied through an external voltage source. Regarding Cr(VI), the remediation mechanism was carried out by four main processes: (i) the reaction of water resulted in a changing the pH of the sample area; (ii) Cr(VI) in soil sample dissolved out continuously during the EKR process; (iii) the dissolved Cr(VI) migrated towards the cathode under the action of an electric field; (iv) the PRB reduced the Cr(VI) by the adsorption and reduction reaction. Generally, Cr(VI) oxyanions migrated from the cathode to the anode under the action of an electric field during EKR and this electromigration process was further improved by the use of additives, i.e. DL-tartaric acid and Tween 80. The DL-tartaric acid controlled the pH of soil through acidification effect and resulted in the release of Cr(VI) ions. But, Tween 80 had good solubility effect which caused the higher desorption of Cr(VI) ions from the soil by forming micelle structures. PRB-1 and PRB-2 had remediated the Cr(VI) through its adsorption and reduction with the presence of CTMAB-Z and Fe(0). The main reactions of Fe(0) and Cr(VI) in the EKR process were as follows:

$$Cr_2O_{72}^- + 2Fe + 14H^+ \rightarrow 2Cr^{3+} + 2Fe^{3+} + 7H_2O$$

$$Cr^{3+} + Fe^{3+} + 6OH^- \rightarrow Fe(OH)3 \downarrow + Cr(OH)3 \downarrow$$

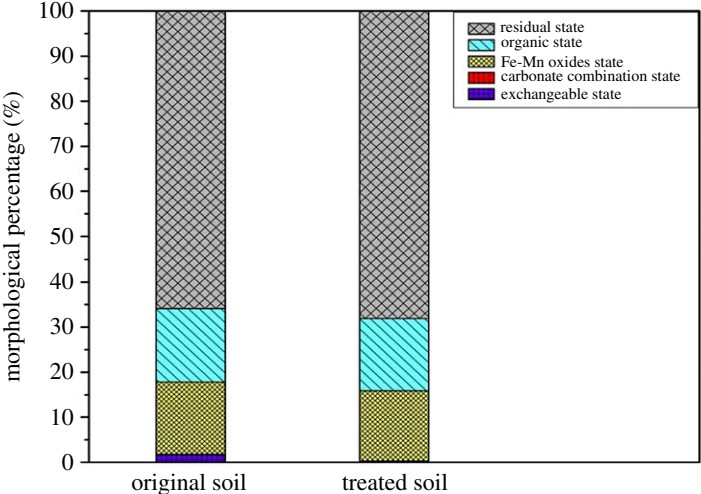

**Figure 15.** Results of the Tessier morphological analysis.

## 4.2. Changes in Cr morphology/forms

The migration effect of Cr varied depending on the morphologies of the soil. The Cr in soil existed as extractable, carbonate, Fe-Mn oxides, organic and residual states. The soil samples before and after the EKR (samples in the A4-S1 region) were subjected to the Tessier extraction method for determination of different morphs/states of Cr. The A4 group, especially the A4-S1 region, achieved the best results in the contrast experiments and was picked up for the following experiments. By conducting the morphological analysis of this typical region (A4-S1), it is reasonable to comprehensively understand the migration of Cr and its effect on the remediation of environmental toxicity with the EKR. According to the results of figure 15, more than 65% of Cr existed as a residual state. Moreover, the easily releasable forms, i.e. exchangeable and carbonate fractions, were significantly decreased after the remediation. The oxides and organic fraction were also decreased due to the acidification effect of DL-tartaric acid. But, the residual fraction increased after remediation which was due to its low release and high stability in the soil. Overall, EKR coupled with PRB-2 in the presence of DL-tartaric acid is a promising and effective approach to reducing the environmental risks caused by Cr exposure.

# 5. Conclusion

In this study, Cr-contaminated soil was treated through EKR coupled with PRB in the presence of chemical additives. The following conclusions were drawn in this study:

(1) In the equilibrium adsorption experiments, the modified zeolite (CTMAB-Z) had higher anionic adsorption capacity when compared with original zeolite (UM-Z) due to the formation of net positive charge on its surface.
(2) In the contrasting experiments, the A4 group having PRB-2 (containing CTMAB-Z and Fe(0)) and DL-tartaric acid possessed the best Cr(VI) remediation effect, as the DL-tartaric acid desorbed the higher amount of Cr(VI) migration towards the anode. Simultaneously, the Cr(VI) was adsorbed and reduced to Cr(III) by CTMAB-Z and Fe(0).
(3) In the multifactor orthogonal experiments, the Cr(VI) remediation efficiency was greatly influenced by the repair time followed by the voltage gradient and the DL-tartaric acid concentration. Moreover, the best remediation efficiency of Cr(VI) was achieved after treating the sample for 8 days at a voltage gradient of 2.5 V cm$^{-1}$ along with 0.15 M of DL-tartaric acid.
(4) Conclusively, the EKR can be effectively used for the removal of easily available forms of Cr(VI), i.e. exchangeable and organic states. But, sole EKR-treated samples after periods of decades can cause a secondary pollution by releasing Cr(VI) ions which existed as an acid extractable form in the soil. Therefore, the EKR coupled with PRB-2 (mixed CTMAB-Z and Fe(0)) in the presence of DL-tartaric acid could be more effective to remediate the Cr-contaminated soils for a long period of time.

Ethics. Our investigation was carried out in full accordance with the ethical guidelines of our research institution and in compliance with Chinese legislation.

Data accessibility. All data and analysis scripts associated with this analysis can be downloaded from https://doi.org/10.5061/dryad.49c3jd7 [39].

Authors' contributions. X.Y. participated in data analysis and the design of the study; X.Y. and F.M. drafted the manuscript text; X.Y., H.L. and Y.Y. participated in the experiments; N.L., X.H., B.J., L.Y. and D.L. supervised the project.

Competing interests. We declare we have no competing interests.

Funding. We received no funding for this study.

Acknowledgements. We thank Miss Fengjiao Xue, Mr Kexiang Liu and Miss Yuewei Zhang for the assistance with experimental equipment installation and guidance in the experimental method, and also thank Dr Tao Huang, Ming Xia and Shan Li for their helpful suggestions.

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
