## [Reviewer comments · Royal Society Open Science]

Review History

RSOS-182138.R0 (Original submission)

Review form: Reviewer 1

Is the manuscript scientifically sound in its present form?

Yes

Are the interpretations and conclusions justified by the results?

Yes

Is the language acceptable?

Yes

Is it clear how to access all supporting data?

Yes

Do you have any ethical concerns with this paper?

No

Have you any concerns about statistical analyses in this paper?

I do not feel qualified to assess the statistics

Recommendation?

Major revision is needed (please make suggestions in comments)

Comments to the Author(s)

Please see attached file (Appendix A).

Review form: Reviewer 2

Is the manuscript scientifically sound in its present form?

Yes

Are the interpretations and conclusions justified by the results?

No

Is the language acceptable?

No

Is it clear how to access all supporting data?

No

Do you have any ethical concerns with this paper?

No

Have you any concerns about statistical analyses in this paper?

No

Recommendation?

Major revision is needed (please make suggestions in comments)

Comments to the Author(s)

1. The manuscript is not well written, many poor sentence construction and grammatical errors. It is strongly recommended that the manuscript is further revised or reviewed by colleagues with English as first language as it is sometimes hard to understand.
2. What type of soil was used in this study? Please classify.
3. Lot of studies have been carryout on Cr contaminated soil and the some more recent literature survey in this area is needed.It is always better to correlate the findings of the study with the reported Literature.
4. The error analysis of test data is needed
5. The removal mechanisms of Cr(VI) is deficiency
6. The Cr morphology/forms is deficiency, other group soil is need

Review form: Reviewer 3

Is the manuscript scientifically sound in its present form?

Yes

Are the interpretations and conclusions justified by the results?

Yes

Is the language acceptable?

Yes

Is it clear how to access all supporting data?

Yes

Do you have any ethical concerns with this paper?

No

Have you any concerns about statistical analyses in this paper?

No

Recommendation?

Accept with minor revision (please list in comments)

Comments to the Author(s)

This research by Xu Yu et al. provides an approach in the effect of chemical additives on EKR of Cr-contaminated soil coupled with a PRB. Remediation technologies of contaminated soil are an important topic and a cutting-edge research direction, especially EKR technology. Although there are several studies in literature that have reported EKR of heavy metal in soil, there is still work to be done regarding the choice of electrolyte solutions, the application of permeable reactive barrier/PRB, the energy-saving of EKR and the mechanism of action of agent, etc. In this study, CTMAB-Z and mixture of CTMAB-Z and Fe (0) were used as PRB reactive media, the effect of chemical additives, such as di-tartaric and tween 80, on EKR of Cr-contaminated soil coupled with PRB, and relatedly reactive mechanisms were discussed. In addition, the three factors including repair time, voltage gradient and dl-tartaric acid concentration were investigated by utilizing multifactor orthogonal experiment, and an optimal reactive condition was disclosed. The manuscript is written clearly and concisely in good English, and it has certain academic value. I would recommend publication of the article if the authors are willing to proceed to a few changes/additions/corrections/ answers indicated below.

P3L43, why do you only add chemical additives into cathode electrolyte? Please add your explanation in manuscript.

P3L56, what does the leaching efficiency refer to? How do you test and calculate? It's not clear, please add relatedly information in your manuscript.

P5L17, Explain why the electrolyte solution in anode compartment became darker with passage of time, and please add your reasons in your manuscript.

P5L21, besides precipitation of heavy metal ions, I believe the tested soil became hardened is mainly because of volatilization of soil moisture caused by thermal effect.

P5L41, the absolutely decrease of ion concentration and quantity in the system may be the main reason, please you rethink the reason why the lowest current was observed in A5 group.

P5L58, please delete or correct the sentence "which indicated that soil was alkaline in nature and had strong pH buffering capacity"; the logic is incorrect, and tested soil properties is not derived from EKR experiments.

P6L6, have you given any thought to the reduction of Cr (VI) by dl-tartaric acid?

P7L20, did you characterize CTMAB-Z supported by Fe (0) after the reaction to verify your

reasoning?

P7L31, why do you choose samples in the A4-S1 region to analyze Cr morphology? Please add your reason in manuscript.

P7L41, please simplify and improve the conclusion, and delete experiment processes and methods which is written in the conclusion.

Decision letter (RSOS-182138.R0)

19-Feb-2019

Dear Professor Li:

Title: Effect of chemical additives on electrokinetic remediation of Cr-contaminated soil coupled with a permeable reactive barrier

Manuscript ID: RSOS-182138

The editor assigned to your manuscript has now received comments from reviewers. We would like you to revise your paper in accordance with the referee and Subject Editor suggestions which can be found below (not including confidential reports to the Editor). Please note this decision does not guarantee eventual acceptance.

Please submit your revised paper before 14-Mar-2019. Please note that the revision deadline will expire at 00.00am on this date. If we do not hear from you within this time then it will be assumed that the paper has been withdrawn. In exceptional circumstances, extensions may be possible if agreed with the Editorial Office in advance. We do not allow multiple rounds of revision so we urge you to make every effort to fully address all of the comments at this stage. If deemed necessary by the Editors, your manuscript will be sent back to one or more of the original reviewers for assessment. If the original reviewers are not available we may invite new reviewers.

RSC Associate Editor:
Comments to the Author:
(There are no comments.)

RSC Subject Editor:
Comments to the Author:
(There are no comments.)

Reviewers' Comments to Author:
Reviewer: 1

Comments to the Author(s)
Please see attached file.

Reviewer: 2

Comments to the Author(s)

1. The manuscript is not well written, many poor sentence construction and grammatical errors. It is strongly recommended that the manuscript is further revised or reviewed by colleagues with English as first language as it is sometimes hard to understand.
2. What type of soil was used in this study? Please classify.
3. Lot of studies have been carryout on Cr contaminated soil and the some more recent literature survey in this area is needed. It is always better to correlate the findings of the study with the reported Literature.
4. The error analysis of test data is needed
5. The removal mechanisms of Cr(VI) is deficiency
6. The Cr morphology/forms is deficiency, other group soil is need

Reviewer: 3

Comments to the Author(s)

This research by Xu Yu et al. provides an approach in the effect of chemical additives on EKR of Cr-contaminated soil coupled with a PRB. Remediation technologies of contaminated soil are an important topic and a cutting-edge research direction, especially EKR technology. Although there are several studies in literature that have reported EKR of heavy metal in soil, there is still work to be done regarding the choice of electrolyte solutions, the application of permeable reactive barrier/PRB, the energy-saving of EKR and the mechanism of action of agent, etc. In this study, CTMAB-Z and mixture of CTMAB-Z and Fe (0) were used as PRB reactive media, the effect of chemical additives, such as di-tartaric and tween 80, on EKR of Cr-contaminated soil coupled

with PRB, and relatedly reactive mechanisms were discussed. In addition, the three factors including repair time, voltage gradient and dl-tartaric acid concentration were investigated by utilizing multifactor orthogonal experiment, and an optimal reactive condition was disclosed. The manuscript is written clearly and concisely in good English, and it has certain academic value. I would recommend publication of the article if the authors are willing to proceed to a few changes/additions/corrections/ answers indicated below.

P3L43, why do you only add chemical additives into cathode electrolyte? Please add your explanation in manuscript.

P3L56, what does the leaching efficiency refer to? How do you test and calculate? It's not clear, please add relatedly information in your manuscript.

P5L17, Explain why the electrolyte solution in anode compartment became darker with passage of time, and please add your reasons in your manuscript.

P5L21, besides precipitation of heavy metal ions, I believe the tested soil became hardened is mainly because of volatilization of soil moisture caused by thermal effect.

P5L41, the absolutely decrease of ion concentration and quantity in the system may be the main reason, please you rethink the reason why the lowest current was observed in A5 group.

P5L58, please delete or correct the sentence "which indicated that soil was alkaline in nature and had strong pH buffering capacity"; the logic is incorrect, and tested soil properties is not derived from EKR experiments.

P6L6, have you given any thought to the reduction of Cr (VI) by dl-tartaric acid?

P7L20, did you characterize CTMAB-Z supported by Fe (0) after the reaction to verify your reasoning?

P7L31, why do you choose samples in the A4-S1 region to analyze Cr morphology? Please add your reason in manuscript.

P7L41, please simplify and improve the conclusion, and delete experiment processes and methods which is written in the conclusion.

Author's Response to Decision Letter for (RSOS-182138.R0)

See Appendix B.

Decision letter (RSOS-182138.R1)

26-Mar-2019

Dear Professor Li:

Title: Effect of chemical additives on electrokinetic remediation of Cr-contaminated soil coupled with a permeable reactive barrier

Manuscript ID: RSOS-182138.R1

It is a pleasure to accept your manuscript in its current form for publication in Royal Society Open Science. The chemistry content of Royal Society Open Science is published in collaboration with the Royal Society of Chemistry.

RSC Associate Editor
Comments to the Author:
(There are no comments.)

Reviewer(s)' Comments to Author:

Appendix A

Reviewer Recommendation and Comments

The manuscript “Effect of chemical additives on electrokinetic remediation of Cr-contaminated soil coupled with a permeable reactive barrier” is clear and follow a logic path. However, before it was accepted for publication in this journal, there are some questions.

1. There were a few similar researches of the remediation of Cr-contaminated soil by EKR-PRB coupled technology. Moreover, the same material (zeolite and zero-valent iron) had also been used in the EKR-PRB remediation of Cr-contaminated soil in the paper entitled “Electro-kinetic remediation of chromium-contaminated soil by a three-dimensional electrode coupled with a permeable reactive barrier. Rsc Advances, 2017. 7(86): 54797-54805”. To distinguish this research from previous studies, authors should add some more information to show the innovation of this research, in the introduction.
2. The specific analytical methods for Cr(VI) concentration in the soil have been missed. And Cr(III) concentration need be obtained during the experiment.
3. Regarding analytical methodologies. Were some validation parameters (e.g. recovery experiments, LOD, LOQ) carried out? The recovery experiments by adding standard Cr samples should be carried out. What was the average recovery ratio?
4. Can authors close the mass balance of total Cr in the experiments? Most likely it is done but should be mentioned in the text. Please check other articles as example. {such as “J.G. Han, K.K. Hong, Y.W. Kim, et al. Enhanced electrokinetic (E/K) remediation on copper contaminated soil by CFW (carbonized foods waste), Journal of Hazardous Materials,177 (2010) 530-538”}

Appendix B

Dear Editor,

Thank you for your letter and reviewer's comments concerning our manuscript entitled "Effect of chemical additives on electrokinetic remediation of Cr-contaminated soil coupled with a permeable reactive barrier" (ID: RSOS-182138).

We have made correction according to comments, which are described below.

Thanks again for your time and letter.

Yours sincerely,

Prof. & Dr. Dongwei Li

College of Resource and Environmental Science, Chongqing University

Chongqing 400044, China

Email: litonwei@cqu.edu.cn

Address: Chongqing University, Chongqing 400044, China

Dear reviewers,

We would like to express our sincere appreciation for your careful reading and helpful comments. Those comments are all valuable and very helpful for revising and improving our paper, as well as the important guiding significance to our researches. We have addressed the points.

Responses to Reviewer #1:

Comment 1: There were a few similar researches of the remediation of Cr-contaminated soil by EKR-PRB coupled technology. Moreover, the same material (zeolite and zero-valent iron) had also been used in the EKR-PRB remediation of Cr-contaminated soil in the paper entitled “Electro-kinetic remediation of chromium-contaminated soil by a three-dimensional electrode coupled with a permeable reactive barrier. *Rsc Advances*, 2017. 7(86): 54797-54805”. To distinguish this research from previous studies, authors should add some more information to show the innovation of this research, in the introduction.

Reply: Thank you for your valuable comments. Although zeolite has been widely used as PRB materials, its adsorption capacity of Cr(VI) anions is limited by its native surface characteristics. These negatively charged zeolite can be further modified with cationic surfactants to form micelle-like bilayer structures, changing the zeta potentials, therefore enhancing the adsorbing capacity of anions. To further improve the adsorption capacity, the zeolite in this study was modified with cetyl trimethyl ammonium bromide (CTMAB), resulting in a better performance. Now, we have added some information to show the innovation of this research in the introduction section according to your guidelines. Now you can see on page (2) at the line (26-35).

Comment 2: The specific analytical methods for Cr(VI) concentration in the soil have been missed. And Cr(III) concentration need be obtained during the experiment.

Reply :(1) we have added the specific analytical methods for Cr(VI) concentration, which can be found in section 3.5(page: 3, line: 39-41; page: 4, line: 1-2).

(2)The concentration of Cr(III) in soil was indirectly determined by measuring the concentration difference between the total Cr and the Cr(VI). The concentration of total Cr in soil was determined by inductively coupled plasma emission spectrometer(ICP-OES). The initial concentration of total Cr and Cr(III) in soil were 36475.32 mg/kg and 35971.32 mg/kg respectively. Now, we presented the data of total Cr and Cr(III) in the two replicates of experiments under the optimum combination, which was shown in the following table.

The concentration of total Cr and Cr(III) under the optimum combination

Group	Total Cr concentration(mg/kg)				Cr(III) concentration(mg/kg)			
	S1	S2	S3	Average	S1	S2	S3	Average
C1	33245.61	33682.81	35246.67	34058.37	33136.39	33570.69	35183.89	33963.66
C2	33751.00	34003.87	34628.43	34127.77	33644.68	33888.84	34556.94	34030.15
Average	33498.31	33843.34	34937.55	34093.07	33390.54	33729.76	34870.42	33996.90

This paper focuses more on the remediation efficiency of Cr(VI) instead of total Cr and Cr(III). From the viewpoint of removal rate, the removal rates of total Cr and Cr(III) are low due to their chemical stability and resistivity of migration. According to the above table, even under the optimal conditions, the removal rates of total Cr and Cr(III) are relatively low, the average removal rate of total Cr and Cr(III) were 6.53% and 5.49% respectively. Luckily, the Cr(III) is stable in the soil and its low toxicity brings less impact on the safety of environment. On the other hand, the concentration of Cr(VI) is critical due to its high toxic, although it only represents a small part of total Cr. The remediation efficiency of Cr(VI) is high as it is easy to migrate with EKR. Overall, this paper discussed the Cr(VI) removal rate and leaching toxicity as the main indicators, but did not consider the two indicators of total Cr and Cr(III).

Comment 3: Regarding analytical methodologies. Were some validation parameters (e.g. recovery experiments, LOD, LOQ) carried out? The recovery experiments by adding standard Cr samples should be carried out. What was the average recovery ratio?

Reply : (1) The LOD in this experiment was 0.007ug/ml, and the LOQ was 0.023 ug/ml.

(2) We used $K_2Cr_2O_7$ and kaolin to prepare the simulated soil with the Cr(VI) concentration of 500 mg/kg. The experiment was carried out under the condition of a voltage gradient of 2.0V/cm, repair time of 5d. The average removal rate of Cr(VI) reached 90%. Therefore, it can be concluded that the EKR has a good effect on the remediation of the simulated soil. Based on this, the actual soil was studied in this paper..

Comment 4: Can authors close the mass balance of total Cr in the experiments? Most likely it is done but should be mentioned in the text. Please check other articles as example. {such as “J.G. Han, K.K. Hong, Y.W. Kim, et al. Enhanced electrokinetic (E/K) remediation on copper contaminated soil by CFW (carbonized foods waste),Journal of Hazardous Materials,177 (2010) 530-538”}.

Reply: In this paper, we used actual Cr-contaminated soil which had a high total Cr content and our research focused on the Cr(VI). After the experiment, the removal rate of Cr(VI) was very high, but more than 90% of total Cr remained in the soil after the experiment. On the other hand, it is difficult to

measure the heavy metal precipitation in the PRB and cathode regions. Therefore, we directly disposed of the waste without analyzing the total Cr in each part. Thanks for your advice, we have found our shortcomings in my current work. We will improve my scientific research level in accordance with your suggestions in future work and make more achievements!

Responses to Reviewer #2:

Comment 1: The manuscript is not well written, many poor sentence construction and grammatical errors. It is strongly recommended that the manuscript is further revised or reviewed by colleagues with English as first language as it is sometimes hard to understand.

Reply: Thank you for your advice. We have made a detailed revision to the manuscript and we hope these will be satisfactory.

Comment 2: What type of soil was used in this study? Please classify.

Reply: The soil was derived from a chemical plant which produced chromium salts. According to the date of XRF(Table 4), the major elements in soil are Ca, O, Si, Cr etc. The Cr(VI) content(504 mg/kg) of the soil is high and the soil is classified as solid waste according to the environmental quality standards for soils (GB15618-2008); According to the amount of sand, the soil is classified as loam; Furthermore, according to the property of pollutants, the soil can be divided into inorganic pollutants - heavy metal contaminated soil.

Comment 3: Lot of studies have been carryout on Cr contaminated soil and the some more recent literature survey in this area is needed.It is always better to correlate the findings of the study with the reported Literature.

Reply: Thank you very much for your advice. We have added more recent references the remediation of Cr-contaminated soil in the introduction(page: 2, line: 11-15, line: 22-26). The references are given below:<https://doi.org/10.1016/j.cej.2017.01.092;>
<https://doi.org/10.1016/j.electacta.2016.06.048;>
<https://doi.org/10.1016/j.jhazmat.2007.03.076;>
<https://doi.org/10.1016/j.jhazmat.2012.08.039;>
<https://pubs.rsc.org/en/content/articlehtml/2017/ra/c7ra10913j>

Comment 4: The error analysis of test data is needed

Rerly: Thank you for your valuable comments. Now, we have added the error analysis for two repeated experiments under optimal conditions. The error analysis of the experiments was shown in Fig. 14.

Thanks for your comments.

Comment 5: The removal mechanisms of Cr(VI) is deficiency.

Reply: We have supplemented the removal mechanisms of Cr(VI) in section 5.1 and we hope these will be satisfactory.

Comment 6 : The Cr morphology/forms is deficiency, other group soil is need.

Reply: we performed morphological analysis on all groups(having tween 80 or mixture) and the parameters of morphology were close to each other. Considering the high performance, A4-S1 was chosen as a typical region for the following experiment. It is reasonable to comprehensively understand the migration of Cr and its effect on the remediation of environmental toxicity with the EKR. The morphological comparison between the original soil sample and the A4-S1 group was also analyzed as the exchangeable and carbonate fractions were decreased significantly and the residual fraction increased after remediation.

Responses to Reviewer #3:

Comment 1 (P3 L43): Why do you only add chemical additives into cathode electrolyte? Please add your explanation in manuscript.

Reply: Thank you for your comments. It was believed that catholyte can improve the remediation effect by controlling pH and enhancing migration effect. We have explained the reason in section 3.4.2(page: 3, line: 27-28) and added some relevant references. The references are given below:

[https://doi.org/10.1016/j.chemosphere.2004.02.033;](https://doi.org/10.1016/j.chemosphere.2004.02.033)

[https://doi.org/10.1016/j.desal.2012.05.023;](https://doi.org/10.1016/j.desal.2012.05.023)

[https://doi.org/10.1016/j.seppur.2011.02.025;](https://doi.org/10.1016/j.seppur.2011.02.025)

[https://doi.org/10.1016/j.envint.2005.05.040;](https://doi.org/10.1016/j.envint.2005.05.040)

Comment 2 (P3 L56) : What does the leaching efficiency refer to? How do you test and calculate? It's not clear, please add relatedly information in your manuscript.

Reply: The leaching toxicity of solid waste refers to the concentration of pollutants in the leaching solution of solid waste obtained through a leaching procedure. The leaching toxicity of Cr(VI) in soil was analysed through the ultraviolet Spectrophotometer after acid solution in accordance with the Chinese standard HJ/T299-2007 solid waste leaching toxicity-sulphuric acid & nitric acid method. The soil was mixed with acid solution(sulphuric acid and nitric acid, pH=3.20±0.05) and shaken at room temperature for 18±2 h at a speed of 120 rpm, then the supernatant after filtering was measured. Now, we have We have explained the leaching efficiency in section 3.5(page: 4, line: 3-8), thanks for your comments.

Comment 3 (P5 L17) : Explain why the electrolyte solution in anode compartment became darker with passage of time, and please add your reasons in your manuscript.

Reply: The electrolytic solution in the anode compartment became darker with a passage of time due to the migration of Cr(VI) from the cathode to the anode under the action of the electric field during EKR. We have added the reason in section 4.3.1(page: 5, line: 18-19). We thank you for this comment.

Comment 4 (P5 L21): Besides precipitation of heavy metal ions, I believe the tested soil became hardened is mainly because of volatilization of soil moisture caused by thermal effect.

Reply: We confessed this point and the modifications have been done according to your valuable comments, which could be seen in section 4.3.1(page: 5, line: 22-23).

Comment 5 (P5L41): The absolutely decrease of ion concentration and quantity in the system may be the main reason, please you rethink the reason why the lowest current was observed in A5 group.

Reply: Thank you for your qualitative comments. The lowest current was observed in A5 group (having 0.1% tween 80), and the low ion concentration in the system was due to less concentration of ions dissolved from the tween 80. We have modified the reason in section 4.3.2 a(page: 5, line: 36-37) according to your guidelines.

Comment 6(P5L58):Please delete or correct the sentence “which indicated that soil was alkaline in nature and had strong pH buffering capacity”; the logic is incorrect, and tested soil properties is not derived from EKR experiments.

Reply: Thank you very much for your valuable suggestion. In spite of higher migration rate of H⁺ than OH⁻, the native alkaline environment of soil with strong buffering capacity caused the partially alkaline pH of the S2 region. Now, we have correct it in section 4.3.2 b(page: 6, line: 6-7) according to your guidelines.

Comment 7(P6L6): Have you given any thought to the reduction of Cr (VI) by dl-tartaric acid?

Reply: Before the EKR experiment, we soaked the soil samples with dl-tartaric acid and detected a large amount of Cr (VI) in the solution, which indicated that the Cr (VI) was released in the soil under the acidification of dl-tartaric acid. Combined with previous studies, it is mainly explained from the acidification of organic acids. Therefore, this paper mainly considered dl-tartaric acid had controlled the pH and acidification effect. The references are given below:

<https://doi.org/10.1016/j.cej.2017.01.092;>

<https://doi.org/10.1016/j.cej.2017.01.092;>

<https://doi.org/10.1016/j.chemosphere.2017.09.104>;

Comment 8(P7L20): Did you characterize CTMAB-Z supported by Fe (0) after the reaction to verify your reasoning?

Reply: After the experiment, SEM analysis of PRB showed that some flocculent particles appeared on the surface of the zeolite, and the experimental results were similar to the reference(<https://link.springer.com/article/10.1007/s11270-016-2790-6>). As this paper mainly discussed the remediation efficiency of Cr(VI) from a macroscopic perspective, the PRB material was not analyzed in detail. In the future experiments, we will focus on the mechanism of PRB through more in-depth experiments.

Comment 9(P7L31): Why do you choose samples in the A4-S1 region to analyze Cr morphology? Please add your reason in manuscript.

Reply: The A4 group, especially A4-S1 region achieved the best results in the contrast experiments and was picked up for the following experiments. By conducting the morphological analysis of this typical region (A4-S1), it is reasonable to comprehensively understand the migration of Cr and its effect on the remediation of environmental toxicity with the EKR. We have added the reason in section 5.2(page: 7, line: 28-31). Thanks for your comments.

Comment 10(P7L41): Please simplify and improve the conclusion, and delete experiment processes and methods which is written in the conclusion.

Reply: The conclusion had been revised in section 6 accordingly. Thanks for your comments.